# A Small Gtp-Binding Protein GhROP3 Interacts with GhGGB Protein and Negatively Regulates Drought Tolerance in Cotton (*Gossypium hirsutum* L.)

**DOI:** 10.3390/plants11121580

**Published:** 2022-06-15

**Authors:** Ziyao Hu, Jianfeng Lei, Peihong Dai, Chao Liu, Abuduweili Wugalihan, Xiaodong Liu, Yue Li

**Affiliations:** 1College of Life Science, Xinjiang Agricultural University, Nongda East Road, Urumqi 830001, China; silvazy201214@126.com (Z.H.); peihong816@163.com (P.D.); liuch_86@126.com (C.L.); xiaodongliu75@aliyun.com (X.L.); 2College of Agronomy, Laboratory of Agricultural Biotechnology, Xinjiang Agricultural University, Nongda East Road, Urumqi 830001, China; kyleijianfeng@163.com (J.L.); wu2021998546@126.com (A.W.); 3Research Center of Cotton Engineering, Ministry of Education, Xinjiang Agricultural University, Nongda East Road, Urumqi 830001, China

**Keywords:** cotton, drought stress, *GhROP3*, *GhGGB*, virus-induced gene silencing (VIGS)

## Abstract

As a plant-specific Rho-like small G protein, the ROP (Rho-related GTPase of plants) protein regulates the growth and development of plants and various stress responses in the form of molecular switches. Drought is a major abiotic stress that limits cotton yield and fiber quality. In this study, virus-induced gene silencing (VIGS) technology was used to analyze the biological function of *GhROP3* in cotton drought stress tolerance. Meanwhile, we used yeast two-hybrid and bimolecular fluorescence complementation assays to examine the interaction between GhROP3 and GhGGB. *GhROP3* has a high expression level in cotton true leaves and roots, and responds to drought, high salt, cold, heat stress, and exogenous abscisic acid (ABA) and auxin (IAA) treatments. Silencing *GhROP3* improved the drought tolerance of cotton. The water loss rates (WLR) of detached leaves significantly reduced in silenced plants. Also, the relative water content (RWC) and total contents of chlorophyll (Chl) and proline (Pro) of leaves after drought stress and the activities of three antioxidant enzymes catalase (CAT), superoxide dismutase (SOD), and peroxidase (POD) significantly increased, whereas the contents of hydrogen peroxide (H_2_O_2_) and malondialdehyde (MDA) significantly reduced. In the leaves of silenced plants, the expression of genes related to ABA synthesis and its related pathway was significantly upregulated, and the expression of decomposition-related *GhCYP707A* gene and genes related to IAA synthesis and its related pathways was significantly downregulated. It indicated that *GhROP3* was a negative regulator of cotton response to drought by participating in the negative regulation of the ABA signaling pathway and the positive regulation of the IAA signaling pathway. Yeast two-hybrid and bimolecular fluorescence complementation assays showed that the GhROP3 protein interacted with the GhGGB protein in vivo and in vitro. This study provided a theoretical basis for the in-depth investigation of the drought resistance–related molecular mechanism of the *GhROP3* gene and the biological function of the *GhGGB* gene.

## 1. Introduction

Cotton (*Gossypium hirsutum* L.) is one of the most important economic crops and plays an important role in the national economy; upland cotton is currently the main cultivar [1]. Cotton is affected by a series of stresses during its growth. Among these, drought is the primary abiotic factor. Drought reduces the morphological and physiological traits and decreases the leaf water potential and sap movement due to alternation of xylem anatomical features in the plants [2]. Different molecular, biochemical, physiological, morphological, and ecological traits and processes of plants are impaired under drought stress conditions [3]. Thus, cotton yield and fiber quality are adversely affected in water deficit environments. In the long-term evolutionary process, plants have developed a set of gene regulation networks for drought adaptation. Elucidating the gene regulatory network might help cultivate new varieties of drought-resistant crops.

Small G proteins are a class of monomeric binding proteins with molecular weights between 20–30 kDa. They are central regulators of many signal transduction processes and are widely present in eukaryotes [4,5]. The small G proteins are typically divided into five families according to their structural and functional differences: Rho, Ras, Arf/Sar, Ran, and Rab. These five families have different divisions and play different roles in cell life activities. Ras and Rho family proteins are signal transduction switches [6,7]. The other three family proteins are involved mainly in regulating vesicle and macromolecular movement [8,9]. The ROP protein is a plant-specific Rho-like small G protein with GTPase activity. It acts as a “molecular switch” in the signal transduction pathway, regulating various life activities of plants [10]. Winge et al. divided the ROP family proteins into two types according to the difference in the amino acid sequences of the ROP protein C-terminal: type I and type II [11]. The ROP protein C-terminal has a canonical CaaL box motif (where C is Cys, a is aliphatic residue, and L is Leu) in which the Cys residue is modified by C20 isoprenyl lipid geranylgeranyl. Type II ROP proteins terminate in a GC-CG box (where G is Gly and C is Cys) in which two Cys residues are separated by five or six aliphatic amino acids [12].

Since Yang et al. cloned the first *ROP* gene in pea [13], ROP proteins have been cloned and identified in a variety of plants. The functional studies have shown that ROP proteins are involved in cell morphological structure, growth, division, subcellular localization, mRNA transport, protein translation, and other life activities [14,15,16,17], mediating the response of plants to biotic [18,19,20,21,22,23] and abiotic stresses [24,25,26,27,28,29,30,31]. However, the investigations of ROP family genes in cotton have focused on fiber development and responses to biotic stresses. *GhRacA* and *GhRacB* were found to be dominantly expressed during cotton fiber initiation and elongation in upland cotton and involved in regulating early cotton fiber development [32]. *GhRac9* and *GhRac13* genes are dominantly expressed during the transition between primary and secondary wall synthesis in cotton fiber development. It was speculated that they might be involved in secondary wall synthesis in cotton [33]. Kim et al. isolated the *GhRAC1* gene from cotton seeds during cotton fiber elongation; it was presumed to regulate fiber elongation by controlling cytoskeleton assembly [34]. Wang used virus-induced gene silencing (VIGS) technology to inhibit the expression of the cotton *GhROP6* gene and significantly reduced the resistance of cotton to *Verticillium* wilt. It was speculated that *GhROP6* was a positive regulator gene in cotton to resist the invasion of *Verticillium dahliae* [35]. Yang et al. reported that the ectopic expression of *GhRac6* in cotton improved *Arabidopsis* resistance to aphids by regulating some aphid resistance indexes [36]. Taken together, current studies on cotton ROP proteins focus on the development of cotton fibers and plant defense response to biotic stresses such as pathogens and herbivorous insects.

However, the function- and stress-related regulatory mechanisms of ROP proteins in cotton in response to drought stress remain unclear. In this study, we analyzed the function of *GhROP3* in drought stress response. The transcript abundance of the *GhROP3* gene changed after drought stress. We therefore speculated that, as an important molecular switch, cotton ROP proteins should regulate various developmental processes and influence plant defense responses to drought stress, which is the most important environmental challenge for plants. To prove this hypothesis, we further investigated the function of the *GhROP3* gene in defense against drought stress in cotton using VIGS technology. Silencing of the *GhROP3* gene in cotton showed that it could improve the tolerance of cotton to drought stress. Yeast two-hybrid and bimolecular fluorescence complementation assays showed that the GhROP3 protein interacted with the GhGGB protein. This study provided a theoretical basis to explore further the molecular mechanism underlying the *GhROP3* gene regulation of drought tolerance in cotton.

## 2. Materials and Methods

### 2.1. Plant Materials and Stress Treatments

Upland cotton “Xinluzao 26” was grown as previously described [37]. When the second true leaf of the plant was fully unfolded, uniformly developed seedlings were collected and subjected to drought (15% polyethylene glycol (PEG) 6000), NaCl (200 mM), room temperature (25 °C, CK), cold stress (4 °C), and heat stress (37 °C) by the methods as described by Ni et al. [38]. The seedlings of water-treated plants served as controls. After treatment for appropriate times (0, 1, 3, 6, 12, and 24 h), the leaves from all the treated and control seedlings were harvested. Meanwhile, some seedlings were soaked in 100 µM ABA and IAA solutions for hormonal treatment. The seedlings treated with a trace absolute ethanol aqueous solution formed the control group. The true leaf tissues were taken after 0, 3, 6, 9, and 12 h. In addition, the roots, stems, leaves, and cotyledons under normal conditions were also collected for RNA isolation and used for tissue-specific expression analysis. The plant leaves at different times and tissues immediately frozen in liquid nitrogen were stored at −80 °C until further use. Each treatment was performed at least three times.

### 2.2. GhROP3 Bioinformatics Analysis

The sequence analysis and multiple alignment were performed using DNAMAN software 8.0. *Gossypium hirsutum* GhRac6 (AYG96716.1), *Arabidopsis thaliana* AtROP1 (NP_190698.1), AtROP2 (NP_173437.1), AtROP3 (NP_179371.1), AtROP4 (NP_177712.1), AtROP9 (NP_194624.1), AtROP10 (NP_566897.1), and AtROP11 (NP_201093.1), *Triticum aestivum* TaRop3 (ADD23345.1), *Oryza sativa* OsRac1 (XP_015621645.1), *Nicotiana tabacum* NtROP1 (CAA10815.2), *Zea mays* ZmROP1 (XP_008644256.1), *Hordeum vulgare* HvRac1 (CAD57743.1), *Musa acuminata* MaROP1 (ABQ15204.1), *Capsicum annuum* CaROP1 (ABB71820.1), and *Eriobotrya japonica* EjROP1 (ACM68949.1) were selected. The phylogenetic tree analysis of GhROP3 and ROP protein sequences from the aforementioned species was performed using MEGA7 (SK, Philadelphia, PA, USA) software.

### 2.3. RNA Isolation and qRT-PCR

Total RNA was isolated from cotton tissues using a Biospin Plant Total RNA Extraction Kit (Bioer, Hangzhou, China). cDNA was synthesized with 2–4 μg total RNA using 5×All-In-One RT MasterMix with AccuRT (ABM, Richmond, BC, Canada).

Quantitative reverse transcription-polymerase chain reaction (qRT-PCR) was conducted with the primers q-GhROP3-F and q-GhROP3-R to analyze the expression patterns of *GhROP3* in cotton seedlings under various treatments and in different tissues and transgenic plants. qRT-PCR was also performed to examine the expression of stress-related genes in *GhROP3*-silenced cotton plants. The cotton *ubiquitin7* gene 37 was used as the standard control. qRT-PCR was performed as described previously [37]. The relative expression levels were calculated using the 2^−ΔΔCT^ formula [39]. All the primers used in this study are listed in Appendix A.

### 2.4. Construction of a Silencing Vector and VIGS in Cotton

With reference to the open reading frame (ORF) sequence (591 bp) of the upland cotton *GhROP3* gene (MK955930), the target sequence (274 bp) for repressing the expression of the target gene was designed using the online software SGN-VIGS (https://vigs.solgenomics.net/ (accessed on 30 December 2014), and PCR amplification was performed. The amplification product was ligated to the TRV2 vector to construct the recombinant vector TRV:GhROP3. Similarly, we selected the chlorophyll synthesis-related *GhCLA1* gene and constructed TRV:GhCLA1 as a visual marker to monitor silencing efficiency [40]. TRV:GhROP3, TRV:GhCLA1, and TRV:RNA1 and TRV:RNA2 vectors were introduced into *Agrobacterium tumefaciens* GV3101 using the freeze-thaw method. The cotton seedlings were grown until both cotyledons were fully expanded and the true leaves just showed their tips. Then, uniformly grown seedlings were selected for VIGS infestation using the injection method. The activation, resuspension, and infection of bacteria were performed by the methods reported earlier [37]. The cotton plants infested with resuspensions containing TRV:GhROP3 and TRV:RNA1, TRV:GhCLA1 and TRV:RNA1, and TRV:RNA2 and TRV:RNA1 were used as the experimental group TRV::GhROP3, the positive control TRV::GhCLA1, and the negative control TRV::00, respectively. About 15 days after infestation, the roots and second true leaves of the experimental, positive control, and negative control plants were sampled. Three biological replicates of each sample were taken for RNA extraction and cDNA synthesis. qRT-PCR was used to detect the efficiency of gene silencing.

### 2.5. Drought Treatment and Determination of Physiological and Biochemical Indicators

*GhROP3*-silenced and negative control plants with consistent growth status were placed in the same-humidity environment and subjected to drought stress by ceasing watering. The photographs were taken before drought treatment, 10 days after drought treatment, and 3 days after rewatering. For the water loss assay with detached leaves, the second true leaves were cut from *GhROP3*-silenced and negative control plants and compared by the weighing method [41]. The fresh weight was recorded every 1 h for a total of 7 h of natural water loss. The second true leaves of *GhROP3*-silenced and negative control plants before and after drought treatment for 10 days were selected. The relative water content (RWC) was measured by the oven drying method as described by Wang et al. [42]. The contents of proline (Pro), hydrogen peroxide (H_2_O_2_), and malondialdehyde (MDA) and the activities of superoxide dismutase (SOD), catalase (CAT), and peroxidase (POD) in the leaves were determined according to the experimental steps mentioned in the instruction manual of the physiological index determination kit (Solarbio, Beijing, China). The total chlorophyll content in plant leaves was determined using a chlorophyll meter (Tupyun, Zhejiang, China). Each sample was repeated three times.

### 2.6. Expression Analysis of ABA and IAA Metabolism- and Signaling Pathway-Related Marker Genes after VIGS Treatment

ABA synthesis-related genes *GhNCED2*, *GhNCED3*, *GhNCED8*, *GhNCED9*, and *GhZEP*; signaling pathway-related genes, as well as *GhABI2*, *GhABI5*, *GhABF2*, and *GhABF4*; catabolism-associated gene *GhCYP707A*; and IAA synthesis−related genes *GhYUC2*, *GhYUC22*, and *GhCYP71A13*; signaling pathway-related genes, as well as *GhARF6*, *GhARF18-1*, and *GhARF18-5* were selected. qRT-PCR was used to analyze the transcript expression changes of the aforementioned genes. The relevant primers used in this study are listed in Appendix A.

### 2.7. Yeast Two-Hybrid Assay

The ORF of *GhROP3* was amplified via PCR, digested, and inserted into a pGBKT7 yeast expression vector to construct a pGBKT7-GhROP3 vector. Similarly, the ORF of *GhGGB* (MK105923) was amplified by PCR, digested, and inserted into a pGADT7 yeast expression vector to construct a pGADT7-GhGGB vector. The preparation, pGBKT7-GhROP3 self-activation detection, co-transformation, and culture of AH109 yeast competent cells were performed by previously described methods [42].

### 2.8. Bimolecular Fluorescence Complementation Experiments

*GhROP3* and *GhGGB* were cloned into the pCambia1300-cEYFP and pCambia1300-nEYFP vectors, respectively. The transformation of *Agrobacterium tumefaciens* EHA105, the co-injection of tobacco leaves, and fluorescence detection were performed by previously described methods [42].

### 2.9. Statistical Analysis

Data were processed and analyzed using GraphPad Prism 8.0 (HM, San Diego, CA, USA) and Excel 2019 (KT, Redmond, WA, USA) software.

## 3. Results

### 3.1. Phylogenetic Analysis of GhROP3

Multiple sequence alignments revealed that the GhROP3 protein contained five I–V effector domain motifs. Motifs I and II were GTPase-binding motifs, and motifs III–V were GTP- or GDP-binding motifs, besides two switch motifs (switch I and switch II), one Rho insertion motif, and the C-terminal variable region (HVR), which were in accordance with the structural features of ROP proteins in plants. The ROP protein C-terminal HVR region had a CAFL motif, conforming to the selectivity of GGB for the amino acid sequence C-terminal of substrate proteins (Figure 1A). The evolutionary tree analysis results showed that GhROP3 belonged to class I ROP proteins and might be involved in prenylation modification reactions, which were speculated to have interactions with GhGGB proteins (Figure 1B).

### 3.2. GhROP3 Protein Interacted with the GhGGB Protein

The *GGB* gene encodes type I protein geranyltransferase, which is involved in protein isoprene modification [43]. Our previous research showed that the last four-amino acid motif of the ROP2 protein in *Arabidopsis* was CAFL, which could be modified by the prenylation of the GGB protein in *Arabidopsis* [44]. The sequence analysis revealed that the last four-amino acid motif of the GhROP3 protein was CAFL, consistent with the substrate characteristics of the prenylation modification of the proteins. Therefore, the yeast two-hybrid assay was used to analyze whether the GhROP3 protein could interact with the GhGGB protein. The yeast colonies transformed with pGBKT7-GhROP3 did not grow on SD/-Trp/-Leu/-His/-Ade plates, indicating that GhROP3 had no self-activating activity. In co-transformation groups, the yeast cells transformed with pGBKT7-GhROP3 and pGADT7-GhGGB could grow on SD/-Trp/-His/-Leu/-Ade plates (Figure 2A). Moreover, BiFC results showed that yellow fluorescence was observed in co-transformed nEYFP-GhGGB and GhROP3-cEYFP and located in the epidermal cell membrane of tobacco, while no fluorescence appeared in the negative control group (Figure 2B). These results showed that the GhROP3 protein interacted with the GhGGB protein.

### 3.3. GhROP3 Was Expressed in Different Tissues of Cotton and Responded to Different Treatments

qRT-PCR was used to analyze the gene expression trends and levels of *GhROP3* under various treatments and in different tissues of cotton. As shown in Figure 3A, the *GhROP3* gene was constitutively expressed in all cotton tissues, but the highest transcript levels were found in the leaves and roots, followed by the stems, and the lowest levels were found in the cotyledons. Under PEG treatment, the expression of *GhROP3* was significantly up-regulated at four time points: 1 h, 6 h, 12 h, and 24 h, with no significant change after 3 h (Figure 3B). At the same time, the results showed that *GhROP3* was involved in responses to salt, cold, heat stress, and ABA, IAA treatments. Under salt treatment, the expression of *GhROP3* was significantly up-regulated at three time points: 3 h, 6 h, and 24 h, with no significant change after 1 h and 24 h (Figure 3C); Under cold treatment, the expression of *GhROP3* was significantly down-regulated change after 1 h and 3 h, and significantly up-regulated at three time points: 6 h, 12 h, and 24 h (Figure 3D); The transcription of the *GhROP3* was induced by heat and IAA treatment and was significantly up-regulated at all time periods after treatment (Figure 3E,G); Under ABA treatment, the expression of GhROP3 was significantly up-regulated change after 3 h, and significantly down-regulated after 6 h and 12 h, with no significant change after 9 h (Figure 3F). The aforementioned results indicated that *GhROP3* was expressed at the transcriptional level in response to drought stress.

### 3.4. Silencing of the GhROP3 Gene Improved Cotton Resistance against Drought Stress

VIGS was used to knock down the transcription of *GhROP3* in upland cotton Xinluzao 26 so as to further explore the role of *GhROP3* in cotton drought resistance. After 15 days of infection, the leaves of positive control *GhCLA1* developed albinism (Figure 4B). qRT-PCR showed that the expression of *GhCLA1* and *GhROP3* in roots and leaves was significantly downregulated. The results indicated that the TRV-VIGS system worked well, and *GhROP3* was successfully silenced in cotton (Figure 4C,D). TRV::00 and TRV::GhROP3 plants were subjected to a natural soil drought stress treatment. No obvious phenotypic differences were found among the TRV::00 and TRV::GhROP3 plants before the stress treatment. After 10 days of natural soil drought, all the plants began to wilt. TRV::00 plants wilted severely and even died. However, TRV::GhROP3 plants displayed less severe wilting of leaves, and their overall health was more optimal (Figure 4E). After rewatering for 3 days, the survival rate of TRV::GhROP3 plants was 52.5%, which was significantly higher than that of TRV::00 plants (35%) (Figure 4F).

In addition, we measured several physiological indicators, including WLR; RWC; contents of Pro, Chl, MDA, and H_2_O_2_; and activities of CAT, POD, and SOD in plants under drought treatment and normal growth conditions. As shown in Figure 4G–O, the WLR from detached leaves was significantly lower in the TRV::GhROP3 plants than in the TRV::00 plants. Under normal conditions, no noticeable differences were found in these physiological parameters between TRV::00 plants and TRV::GhROP3 plants. After drought stress, the RWC, contents of Pro and Chl, and activities of CAT, POD, and SOD were significantly higher in the TRV::GhROP3 plants than in the TRV::00 plants. However, the contents of H_2_O_2_ and MDA in the TRV::GhROP3 plants were significantly lowered compared with those in the TRV::00 plants. These results indicated that the knockdown of the *GhROP3* gene could enhance the tolerance of cotton plants to drought stress.

### 3.5. GhROP3 Negatively Regulated ABA Signaling Pathway, and Positively Regulated IAA Signaling Pathway

We investigated the expression changes of several genes associated with ABA, IAA metabolism, and signaling pathways to explore the drought resistance mechanism underlying the enhanced drought tolerance in the *GhROP3-*silenced plants. The qRT-PCR results are illustrated in Figure 5. Five ABA biosynthesis-related genes (*GhNCED2*, *GhNCED3*, *GhNCED8*, *GhNCED9*, and *GhZEP*) and four ABA signaling pathway-related genes (*GhABI2*, *GhABI5*, *GhABF2*, and *GhABF4*) were significantly upregulated, but the catabolism-associated *GhCYP707A* gene was downregulated in TRV::GhROP3 plants. Moreover, three IAA biosynthesis-related genes (*GhYUC2*, *GhYUC22*, and *GhCYP71A13*) and three IAA signaling pathway-related genes (*GhARF6*, *GhARF18-1*, and *GhARF18-5*) were significantly downregulated. The results suggested that the silencing of *GhROP3* increased the mRNA levels of ABA synthesis- and signal response-related genes and reduced the expression levels of IAA synthesis- and signal response-related genes, enhancing the plant resistance to drought stress.

## 4. Discussion

### 4.1. GhROP3 Is a Prenylation Modification Substrate of GhGGB Protein

Protein prenylation is required for protein-protein and protein-membrane interactions, and is a form of post-translational modification of proteins. The acyl group is covalently attached to the cysteine of the last four amino acids CaaX at the carbon end of the protein (C refers to Cys, which is usually an aliphatic amino acid, and X is usually Met, Ala, Gln, Ser, or Cys) [45]. Prenylation is executed by three protein complexes: protein farnesyl transferase (PFT), type I protein geranyltransferase (PGGT I), and type II protein geranyltransferase (PGGT II). Among these, PFT and PGGT I are composed of α and β subunits. The two enzymes share one α subunit, PLP, but the β subunits are different, namely ERA1 and GGB [46]. *Arabidopsis* has 11 ROP proteins. Among these, the C-termini of AtROP1-AtROP8 proteins carry the CaaL sequence, which is characteristic of a type I ROP protein. The β subunits of farnesyltransferase and type I geranyltransferase, encoded by the ERA1 and GGB genes, respectively, catalyze the protein prenylation modification process [47]. The cotton type I ROP protein GhRAC13 with CAFL in its C-terminal is prenylated in vitro [48]. The sequence alignment showed that the C-terminal sequence of the GhROP3 protein was CAFL (Figure 1A) and belonged to class I ROP protein (Figure 1B), suggesting that it might be the modified substrate of the GhGGB protein. The results of yeast two-hybrid and BiFC assays showed that the GhROP3 protein interacted with the GhGGB protein both in vitro and in vivo (Figure 2A,B), indicating that GhROP3 was a target protein for GhGGB prenylation, which was consistent with the *Arabidopsis* ROP2 protein with CAFL in its C-terminal [44].

### 4.2. GhROP3 Played a Negative Role in the Response of Plants to Drought Stress

Previous studies showed that the expression pattern of a gene was usually an indicator of its function [49]. The transcription of *GhROP3* was affected by drought stress (Figure 3B), and also responds to abiotic stresses such as salt, cold and heat stresses (Figure 3C–E), suggesting that *GhROP3* might act as a regulator connecting multiple signaling networks in the process of plant adaptation to abiotic stress. However, the expression patterns under different stresses were different to a certain extent, and it was speculated that its functions in the stress response of cotton to different stresses was different. *GhROP3* was silenced in cotton using VIGS technology to confirm the function of *GhROP3* when plants were under drought stress. As shown in Figure 4E, *GhROP3-*silenced plants significantly enhanced the drought tolerance of cotton plants. Drought stress could lead to changes in a series of physiological indicators in plants [50]. Hence, we measured physiological parameters to evaluate the function of *GhROP3* in the drought stress response. *GhROP3*-silenced plants showed a significantly lower WLR of detached leaves in cotton (Figure 4G) and a markedly higher RWC after 10 days of drought stress (Figure 4H). As expected, *GhROP3*-silenced plants showed better water retention compared with control plants.

Drought stress disturbs plant photosynthesis, reduces chlorophyll content, and causes damage to the photosynthetic apparatus, ultimately leading to oxidative stress and the formation of reactive oxygen species (ROS) [51]. If toxic radicals are not quickly removed or inactivated by the antioxidant defense system, they can affect plant growth and yields as a result of intensified MDA production, protein degradation, DNA breakdown, and perturbation of cell metabolism [52]. The plants produce antioxidants, flavonoids, and secondary metabolites that play a role in detoxifying ROS, protecting the plants against abnormal conditions (i.e., stress), and promoting protein and amino acid stabilization to cope with drought-mediated oxidative stress [53]. The normal physiological metabolism is maintained by removing redundant ROS using some antioxidant enzymes (CAT, POD, SOD, and so on) in plants [54]. Studies showed that a low content of MDA and ROS was closely related to drought and salt tolerance regulated by ROPs in different plants [28,29,30]. This study also found that after 10 days of drought stress, the *GhROP3*-silenced plants had lower contents of H_2_O_2_ and MDA in leaves compared with the control plants; also, the activities of three antioxidant enzymes CAT, POD, and SOD significantly increased (Figure 4J–L,N,O). Proline, acting as an osmo-compatible solute and a nonenzymatic antioxidant, can help decrease the cell osmotic potential under stress conditions and stabilize proteins by preserving their chemical structure [53]. Under drought stress, plants can accumulate a certain amount of Pro to improve their osmotic regulation ability and their own stress tolerance [55]. After drought stress, the Pro content in the leaves of *GhROP3*-silenced plants significantly increased (Figure 4I), indicating that the silencing of the *GhROP3* gene improved the osmotic regulation ability of plants. Drought stress induces stomatal closure in plant leaves and thus inhibits the photosynthetic activity of chloroplasts. At the same time, under the influence of drought stress, the decomposition rate of chlorophyll in the plant itself tends to increase, whereas the synthesis rate decreases significantly, resulting in a decreasing trend of its content [56,57]. After drought stress, the total chlorophyll content in the leaves of the silenced plants was significantly higher than that in the control plants (Figure 4M), indicating that, after suppressing the expression of the *GhROP3* gene, the silenced plants under drought stress had better light and nutritional and health statuses than the control plants. The aforementioned results preliminarily indicated that the *GhROP3* gene acted as a negative regulator of cotton drought tolerance by accumulating related osmotic regulators and nutrient pigments, increasing the activities of related antioxidant enzymes, and reducing the accumulation of ROS and related oxidative products.

### 4.3. GhROP3 Mediated Drought Tolerance in Cotton by Participating in the Regulation of ABA and IAA Signaling Pathways

Plants have evolved various regulatory mechanisms to adapt to stresses such as drought, and hormone regulation is one of the important pathways [58]. Generally, when plants are exposed to drought, the level of drought-related hormones, especially ABA, increased. The ABA hormone regulation is one of the important pathways. Moreover, the higher the drought intensity, the higher the ABA concentration [59]. ROP1 was found to mediate the ABA signaling pathway as a negative regulator [60]. Subsequently, ROP2, ROP6, ROP10, and ROP11 were found to negatively regulate ABA-controlled seed germination or stomatal closure, root elongation, and expression of related genes [24,25,61,62]. The ROP protein acts as a molecular switch in the signal transduction pathway, and the transcript of the *GhROP3* gene responds to exogenous ABA treatment in cotton (Figure 3F), indicating that it might be involved in the ABA signaling pathway by regulating the expression of the related genes. Zeaxanthin epoxidase (ZEP) and 9-*cis*-epoxycarotenoid dioxygenase (NCED) are two key enzymes participating in the ABA synthesis pathway. The expression of the two enzymes is significantly upregulated, and the level of endogenous ABA increases after exposure to drought, which finally improves plant stress resistance [61]. Many studies have shown that the NCED family genes act as a positive regulator in the drought tolerance response in plants [63,64,65]. In this study, the expression of ABA biosynthesis-related genes *GhNCED2*, *GhNCED3*, *GhNCDE8*, *GhNCED9*, and *GhZEP* was upregulated in the *GhROP3*-silenced plants (Figure 5A). Furthermore, the expression of genes related to the ABA signaling pathway, including *GhABI2*, *GhABI5*, *GhABF2*, and *GhABF4*, was significantly upregulated after the silencing of *GhROP3* (Figure 5A). As a key enzyme of ABA catabolism. CYP707A regulates the ABA content in plants under stress via the ABA 8′-hydroxylation pathway [66,67]. The expression of *GhCYP707A* significantly decreased after the silencing of *GhROP3* (Figure 5A). These results suggested that the *GhROP3* gene mediated drought tolerance by participating in the negative regulation of the ABA signaling pathway in cotton.

Auxin acts as a central organization hub in controlling plant growth and development. It regulates a series of life activities, such as apical dominance, organogenesis, and reproductive development of plants by affecting cell growth, differentiation, and structure [68]. Auxin also influences plant resistance to stress. Plants can flexibly regulate each stage of the auxin biosynthesis pathway so that the local auxin content fluctuates within a certain range and activates or inhibits the expression of a variety of stress resistance-related genes [69]. The YUCCA (YUC) genes encode flavin monooxygenases involved in auxin biosynthesis [70]. *GhYUC22* downregulation reduced the sensitivity of cotton to drought by reducing the content of IAA and increasing the content of ABA and the expression of related genes [42]. Auxin/Indoleacetic acid (Aux/IAA) proteins repress auxin-responsive transcription through inactivating ARF function [71]. *OsIAA18* and *OsIAA20* play a positive role in drought by regulating stress-induced ABA signaling [72,73], indicating that auxin and its signaling negatively regulate drought tolerance. ROP positively regulates IAA and is an important branch of the noncanonical auxin signaling pathway [74,75,76], which is consistent with the observation that the expression of IAA biosynthesis- and signaling pathway-related genes, such as *GhYUC2*, *GhYUC22*, *GhCYP71A13*, *GhARF6*, *GhARF18-1*, and *GhARF18-5*, was significantly downregulated after the silencing of *GhROP3* (Figure 5B).

## 5. Conclusions

In this study, reverse genetics approaches were employed to explore a novel biological function and potential mechanism of ROP genes in drought tolerance of cotton. Our results indicated that *GhROP3* functioned negatively in response to drought stress and was a potential candidate gene for developing drought resistance in cotton mutants detected by genome-editing technologies. This study provided not only data for exploring the molecular mechanism underlying cotton drought tolerance but also a rationale for breeding novel cotton germplasm resistant to drought stress.

## Figures and Tables

**Figure 1 plants-11-01580-f001:**
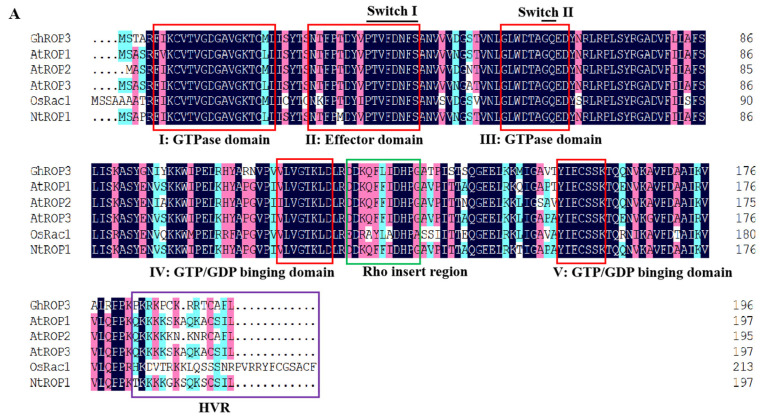
GhROP3 sequence analysis. (**A**) Multiple sequence alignment. (**B**) Phylogenetic tree analysis. Red star marks the target protein GhROP3.

**Figure 2 plants-11-01580-f002:**
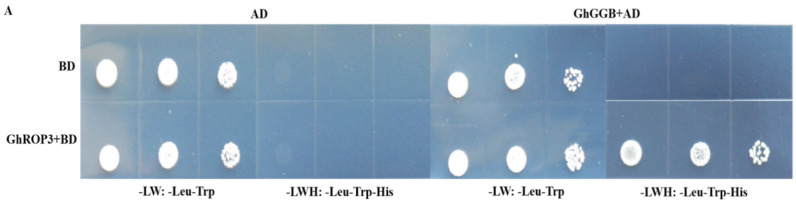
GhROP3 protein interacted with the GhGGB protein. (**A**) Yeast two-hybrid assay. Negative control: pGADT7 + pGBKT7, pGBKT7-GhROP3 + pGADT7, and pGBKT7 + pGADT7-GhGGB. Trp, Tryptophan; Leu, leucine; His, histidine; Ade, adenosine. (**B**) Bimolecular fluorescence complementation. Negative control: nEYFP + cEYFP, nEYFP-GhGGB + cEYFP, and cEYFP-GhROP3 + nEYFP. UV: YFP fluorescence; CHI: RFP fluorescence; DIC: light; Merged: superimposed.

**Figure 3 plants-11-01580-f003:**
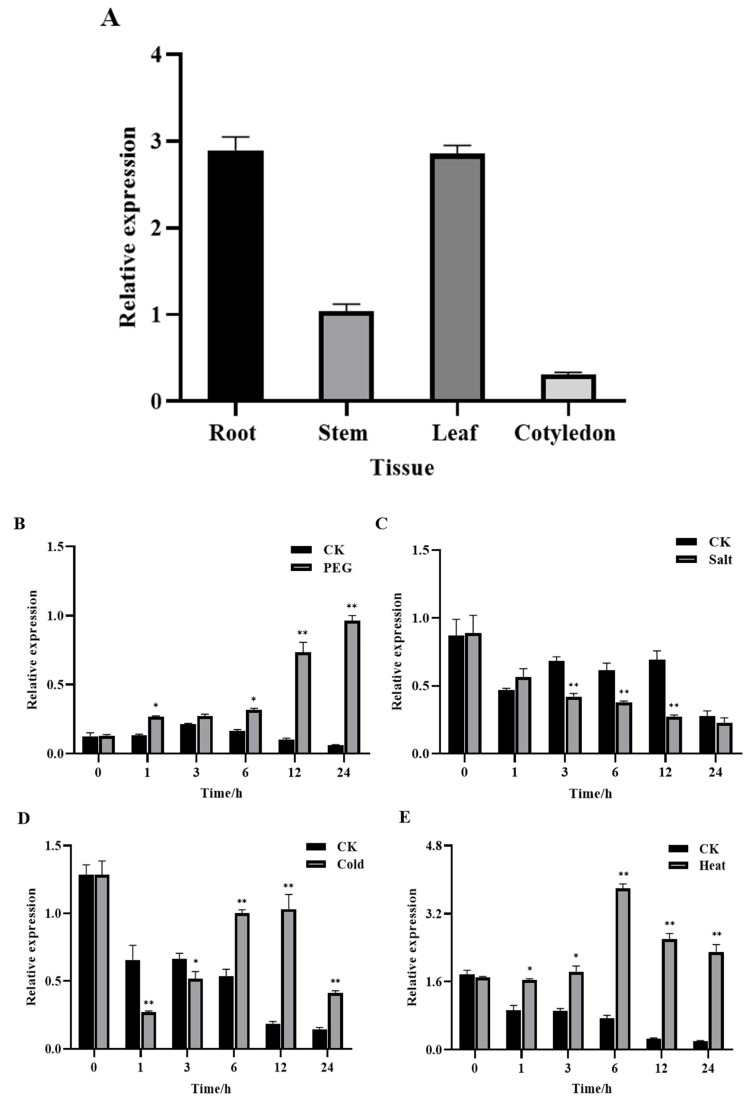
Transcript levels of the *GhROP3* gene. (**A**) Transcript levels of *GhROP3* in cotton roots, stems, leaves, and cotyledons. Transcript levels of *GhROP3* in leaves under abiotic stress treatments, including treatments with polyethyleneglycol (PEG) (**B**), NaCl (**C**), cold (**D**), heat (**E**), abscisic acid (ABA) (**F**), and auxin (IAA) (**G**). Statistical analyses were performed using the Student *t* test. Asterisks indicate a significant difference compared with the control. *, *p* < 0.05; **, *p* < 0.01; ***, *p* < 0.001.

**Figure 4 plants-11-01580-f004:**
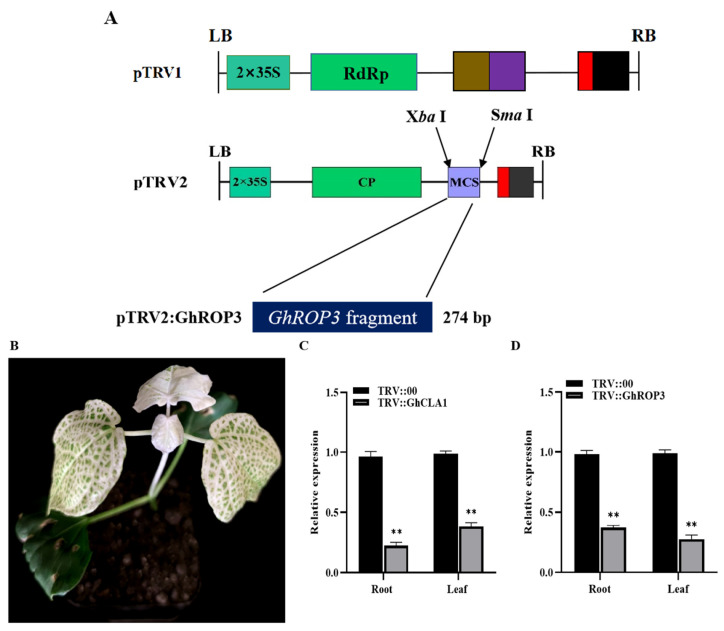
Cotton plants in which the *GhROP3* gene was silenced were more sensitive to drought stress compared with negative control plants. (**A**) TRV1 and TRV2:GhROP3 vector diagram. (**B**) Phenotype of silenced *GhCLA1* in cotton. (**C**,**D**) Silencing efficiency test of *GhCLA1* and *GhROP3*. (**E**) Phenotypes of *GhROP3*-silenced and negative control plants before and after drought stress and after rehydration. (**F**) Survival rate of the negative control plants and *GhROP3*-silenced plants after rewatering. Determination of drought-related biochemical indexes, including water loss rate (WLR) (**G**), relative water content (RWC) (**H**), proline (Pro) (**I**), catalase (CAT) (**J**), peroxidase (POD) (**K**), superoxide dismutase (SOD) (**L**), chlorophyll (Chl) (**M**), hydrogen peroxide (H_2_O_2_) (**N**), and malondialdehyde (MDA) (**O**). Statistical analyses were performed using the Student *t* test. Asterisks indicate a significant difference compared with the control. *, *p* < 0.05; **, *p* < 0.01.

**Figure 5 plants-11-01580-f005:**
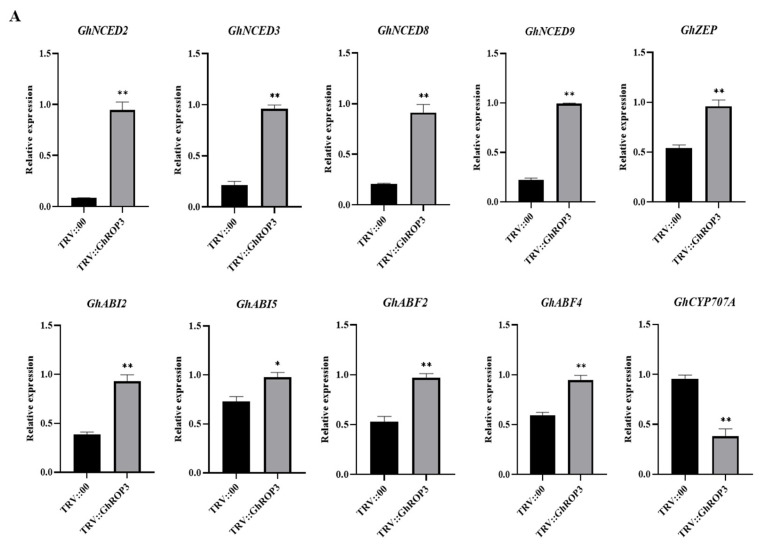
Analysis of the expression of genes related to ABA and IAA synthesis in true leaves of *GhROP3*-silenced and negative control plants. (**A**) ABA synthesis- and signaling pathway-related genes. (**B**) IAA synthesis- and signaling pathway-related genes. Statistical analyses were performed using the Student *t* test. Asterisks indicate a significant difference compared with the control. *, *p* < 0.05; **, *p* < 0.01.

## Data Availability

The raw sequence data reported in this study have been deposited in the National Center for Biotechnology Information sequence read archive (*GhROP3* accession: MK955930 and *GhGGB* accession: MK105923).

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
