# Peer review of "A Small Gtp-Binding Protein GhROP3 Interacts with GhGGB Protein and Negatively Regulates Drought Tolerance in Cotton (*Gossypium hirsutum* L.)"

_plants, 2022, doi:10.3390/plants11121580_

Round 1

Reviewer 1 Report

General comments

I have read the manuscript (plants-1730298). Entitle: A small GTP-binding protein GhROP3 interacts with GhGGB protein and negatively regulates drought tolerance in cotton (Gossypium hirsutum L.) written by Ziyao Hu et. al., for publication of plants MDPI. In this study, author used virus-induced gene silencing (VIGS) technology was used the biological function of GhROP3 in cotton drought stress tolerance. Author found that the water loss rates of detached leaves were significantly reduced in silenced plants. Also, the relative water content (RWC) and total chlorophyll (Chl) and proline (Pro) contents of leaves after drought stress and the activities of CAT, SOD, POD significantly increased, and the contents of hydrogen peroxide (H2O2) and malondialdehyde (MDA) significantly reduced.

The overall research is well conducted, and research is obvious application potential for the readers because this study provided a theoretical basis for the in-depth investigation of the drought resistance-related molecular mechanism of the GhROP3 gene and the 30 biological function of the GhGGB gene. In this sense, the manuscript is much valuable. However, I found some points, especially the flow of the text and lack of potential references, and lack of connection of story in different paragraphs, especially in the introduction and discussion sections. The author should provide enough examples and their interpretation of different traits of biochemical responses by the latest and appropriate references, some of which I mentioned below. Overall after I evaluate and request the author for this manuscript as a “MAJOR REVISION”.

 Major suggestions

1) Introduction: The introduction is well starting with the economic importance of cotton which is much appreciated. The other paragraph is well connected by describing the gene level in different crops. However, author goal is also related to the stress-related regulatory mechanisms. Overall, in the introduction I was unable to find the negative consequence or its negative effect in plant biology.  the author should mention the overall effect of drought in the introduction. Please read and cite this as a reference. Entitle: Entitle “Response of drought stress in prunus sargentii and larix kaempferii ...https://doi.org/10.1016/j.foreco.2020.118099” Please mentioned that “drought reduced the morphological and physiological traits, reduce the leaf water potential and sap movement due to alternation of xylem anatomical features in the plants”. Then only the author should descript the other’s abiotic stress.

2) Hypothesis and objectives of the study: Author should rephrase the text more deeply about the hypothesis of the study and at the same time author should connect the objectives. In this study, author study the GhROP3 in the drought stress response. Silencing of the GhROP3 gene in cotton showed that it could improve tolerance. Please revise the text from lines no 77-83 write more logically by including both parts. The hypothesis of the study is an important thing, and it gives another strength to the introduction. The hypothesis should be very clear in the introduction sections because, without appropriate literature, questions, or hypotheses in the introduction section the entire text will be unclear. The author should give special attention and the sequential presentation of the content in the introduction with presenting the hypothesis of the study.

3) Discussion: Author should Improve discussion more logically with clearer. Author should include somewhere in a discussion about the antioxidant such as CAT and POD and secondary metabolites under drought stress condition and why formation the ROS under drought? Refer to these two articles for better clarify (1) https://doi.org/10.1038/s41598-019-55889 (2) https://doi.org/10.1016/j.scitotenv.2021.146466 and mention somewhere in that paragraph “abiotic stress especially environmental stress (I.e. drought) plant produces the ROS when these plant exposed to the stress condition and plant produce antioxidant, flavonoids, and secondary metabolites play to the role for protecting the plant for detoxifying ROS and protect the plant to protect the abnormal condition (i.e. stress) and protein and amino acid stabilization”.

 Other comments

4) Line no. 380 (Discussion): Drought-related hormones, especially abscisic acid (ABA). Generally, when plants are exposed to the drought, the level of ABA is increased, moreover, the higher the drought intensity higher the ABA concentration. Which is provide the signal from the soil to the root and plant system thoroughly. The ABA hormone regulation is one of the important pathways. The text related to the ABA under different drought intensities is well described in this article https://DOI:10.1016/j.scienta.2018.11.021. Please refer to this article in line no. 380.

5) Line no. 426 (Conclusion): The conclusion for me comes off as repetitive of the abstract or a summary of the results section. I would love to read striking points and take-home messages that will linger in the readers’ minds. What is the novelty, how does the study elucidate some questions in this field, and the contributions the paper may offer to the scientific community?

 6) Line no. 446 (Reference): please double-check the citations, their style, spell check, and other grammatical errors. moreover, I request to the authors for revision throughout the manuscript according to the journal rules.

Good Luck!

Author Response

请见附件,感谢您的宝贵意见。

Reviewer 2 Report

Hu et al present the importance of the G protein ROP in the regulation of cotton tolerance to drought. They use several approaches and technologies in order to prove the significance of this protein in drought regulation. They first use the VIGS technology for gene silencing in order to show the importance of this protein in plant survival under stress. Then they use a two hybrid assay in order to identify interacting candidates as well as BFC to test the interaction between GhROP3 and GhGGB. Following several markers they conclude that GhROP3 is a negative regulator of cotton response to drought by negatively regulating ABA signalling and positively regulating IAA signalling. They also show that a type I geranyltransferase protein GGB interacts with GhROP3 and could be modified by the prenylation of GhGGB a modification required for protein-protein interaction.

The experimental strategy of this work is sound and well organised. However I fill that the proof of the GhROP3 prenylation is missing and this could be easily proved by an MS analysis. Also the authors should check the references since a lot of them appear mistaken.

Author Response

Please see the attachment, thank you for your valuable advice.

Round 2

Reviewer 1 Report

Dear Author

I have read the revised manuscript (plants -1730298). Entitle: A small GTP-binding protein GhROP3 interacts with GhGGB protein and negatively regulates drought tolerance in cotton (Gossypium hirsutum L.) for publication in plants MDPI. This is the second submission made by the author. The author addressed all the questions and suggestions that I raised the issue in the review of the original manuscript. I satisfy the author’s revisions throughout the paper. Especially author improved the introduction and discussion section very well inflow. Now, this manuscript improved the flow of writing, which was comparatively shallow in the original version but in this revised copy author addressed all the quarries and suggestions very well. Before accepting this manuscript if there is anything needed to be revised by the author, especially English grammar, or spell check, I request this manuscript is currently in “Minor Revision” and any grammatical error author may improve in this stage. Thank you.

Author Response

Dear Reviewer,

We have checked and corrected the grammar and spelling of words in the manuscript according to your request, Thank you very much for your valuable advice.